# Corporate Social Responsibility on Twitter: A Review of Topics and Digital Communication Strategies' Success Factors

Katharina Pilgrim *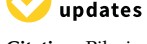 and Sabine Bohnet-Joschko 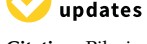

Management and Innovation in Health Care, Faculty of Management, Economics and Society,
Witten/Herdecke University, Alfred-Herrhausen-Str. 50, 58455 Witten, Germany
* Correspondence: katharina.pilgrim@uni-wh.de; Tel.: +49-2302926595

**Abstract:** Corporate social responsibility (CSR) has become increasingly important for companies in recent years. On the one hand, regulatory frameworks require the disclosure of measures for sustainable management. On the other hand, for long-term corporate success, stakeholders must be strategically engaged in the dialog on sustainability aspects. Social media and Twitter in particular offer the potential to foster a meaningful stakeholder dialogue on CSR topics. Twitter's strategic realignment due to Elon Musk's acquisition in the fall of 2022, provides an opportunity to capture research results on activities and strategies on the platform systematically, and to synthesize information for future comparative longitudinal studies of changes in usage. We conducted a literature review including 42 papers to contribute to the body of evidence on CSR communication strategies on Twitter across industries and countries, by deriving interdisciplinary suggestions for strategic CSR-related stakeholder management. Results cover relevant CSR topics, prioritized stakeholder groups for CSR communication on Twitter, and successful communication strategies for companies to obtain beneficial results, such as generating social media capital. The results contribute to the strategic planning and implementation of CSR stakeholder management on Twitter, and offer starting points for future studies on social-media mining and CSR communication strategies.

**Keywords:** corporate social responsibility; Twitter; stakeholder management; social media communication; social media; CSR; communication strategy

## 1. Introduction

As corporate social responsibility (CSR) concepts continue to evolve, balancing differing stakeholder interests remains a persistent management challenge [1]. Starting at a voluntary level, international legislators are now putting pressure on companies by making the implementation of CSR measures in defined areas and disclosure, mandatory [2]. At the same time, to enhance stakeholders' company perception, its actions and ethics [3], CSR disclosure is becoming a more sophisticated and strategically motivated process with expectations of a return [4]. Thus, carrying out socially responsible behavior becomes a strategy of legitimation and survival [5,6]. As sharing nonfinancial information can ameliorate information asymmetries and leave a company in a better position compared to its non-disclosing competitors [7], the utilization of communication channels beyond mandatory reports, dedicated primarily to regulatory authorities, to address other relevant stakeholders. is key. From 2024 onwards, due to the new Corporate Sustainability Reporting Directive (CSRD) over 20,000 companies listed on the European market will be required to implement and report on corporate social responsibility (CSR) activities for the first time. Knowing which activities work best for stakeholders or within one's industry and should be picked up, addressed, and reported on preferentially, becomes a key challenge. Utilizing existing publicly available information from sources with CSR topic engagement can provide a powerful solution [8]. In this context, social media platforms offer large data on public CSR discussions, which can serve as an information resource for the development

of a company's own CSR strategy. At the same time, trust, authenticity, and transparent communication of information describe existing challenges of communication on social networks with respect to corporate social responsibility [9–11]. Successfully deployed social media can thus be a powerful tool for building sustainable corporate communication. To generate beneficial outcomes, companies need to have answers to the following questions: which CSR-related content should be communicated? Which stakeholders should be preferentially addressed? Which communication techniques should be used to leverage stakeholder management-outcomes (in order not to be perceived as greenwashing or social washing)?

In the search for suitable communication channels, Twitter in particular offers a good environment for corporate communication and stakeholder management in the context of CSR [12]. The microblog became a medium for large user-interaction on ethical corporate practices [13] and a valuable channel for creating an emotional bond, positively impacting consumer trust [14]. At the same time, Twitter offers low-threshold access to historical data via the publicly available API. Thus, the number of studies on CSR communication on Twitter, as a sentiment and opinion-forming platform, exceeds those that include other business networks, such as LinkedIn, by approximately three times (Google Scholar hits of CSR and Twitter: 208,000 in August 2022 compared to 76,000 hits of CSR and LinkedIn, see scholar.google.de).

Twitter's strategic realignment, due to Elon Musk's acquisition in the fall of 2022, provides an opportunity to capture research results on activities and strategies on the platform systematically and to synthesize information for future comparative longitudinal studies of changes in usage.

As the amount of literature engaging in the specific field of CSR communication on Twitter is limited, the body of evidence becomes unsuitable for a bibliometric analysis (e.g., conducting a network analysis) to assess the social and structural relationships between different research fields [15]. Scoping reviews in this context have become an increasingly popular approach for synthesizing research evidence by mapping the existing literature in the field of interest in terms of the volume, nature, and characteristics of the primary research [16]. Our scoping review thus aims to identify knowledge gaps, scope a body of literature and clarify concepts [17] by identifying (1) parameters that define relevant content in the context of digital CSR stakeholder management and (2) favorable communication techniques that are a driver for a successful CSR communication-strategy on Twitter. While existing reviews regarding CSR on Twitter focus on trends and topic changes over time [18], our review combines, compares and discusses the recent literature, including those reviews focusing on CSR topics, and adds findings regarding communication strategies to condense research findings and elaborate on the importance of implementing a proactive and meaningful stakeholder management-strategy for CSR on Twitter. This review aims to contribute to the body of evidence on CSR communication-strategies on Twitter across industries and countries, by deriving interdisciplinary suggestions for strategic CSR-related stakeholder management. The review results contribute to a better understanding of CSR communication and of how strategic stakeholder-management on Twitter can add value, especially for those companies who have to prepare a sustainability report for the first time, from 2024 onwards.

The paper is structured as follows: initially, we provide background information on "Digital CSR communication strategies and stakeholder-management". Section 2 describes the systematic research process including data collection and analysis. In the "Results and Discussion" section we first provide sample descriptives by identifying the geographical focus of the conducted research (scope), the addressed and included stakeholder-groups, and the time lapse of the research topics (aims). We then present the identified and clustered CSR-related topics which are communicated and the trends, to gain a better understanding of which topics seem to be suitable for digital CSR-related communication for stakeholder-management on Twitter. We discuss findings on identified CSR communication-strategies regarding strategy choice and the factors for successful digital stakeholder-management.

Finally, we point out implications for digital CSR related stakeholder-management in practice, shed light on limitations and future research, and close with a conclusion.

## 2. Digital CSR Communication-Strategies and Stakeholder-Management

The stakeholder concept, also referred to as the stakeholder approach, is a theory by Edward Freeman, according to which a company, in the course of its strategic orientation, with the aim of maximizing capital, must first define relevant stakeholders to secondly analyze them, to develop corporate goals that are in line with all stakeholder interests [19]. In 1963, the term was first used in an international memorandum by the Stanford Research Institute (SRI), in which stakeholder was used to describe all key groups without which a company could not exist [20]. In contrast, strategic management in the shareholder value approach focuses exclusively on the needs and monetary objectives of the shareholders (owners of the company via shares) [21]. In Freeman's stakeholder approach, the company's goals are optimally defined, taking into account the interests of all stakeholders (including shareholders), so that they can be jointly pursued and supported [5]. Accordingly, the company considers its entire socio-economic context to ensure long-term success. Dynamic changes in the individual relationships' relevance over the course of corporate development must be taken into account when prioritizing stakeholders [19]. The roots of these dynamics in an economy and on markets were already recognized by Schumpeter (1931) [22] and categorized by Porter (1979) [23], in his five-forces model. Here, the five forces determine the profit structure of an industry by determining how the economic value it creates is allocated [23]. This value can, of course, be siphoned off by rivalry among existing competitors, but it can also be traded away by the power of suppliers or the power of customers (both external stakeholders, according to Freeman) or constrained by the threat of new entrants or the threat of substitutes [23]. From a system perspective, stakeholders can be assigned to the direct and indirect environment, measured by their degree of influence and impact [19]. Accordingly, forces act on the company which not only influence its own long-term objectives, but also its short-term positioning with regard to external presentation and impact, under which corporate social responsibility is to be classified. In this context the legitimacy theory states that organizations continuously try to ensure that they carry out activities in accordance with societal boundaries and norms [24], which results in voluntary disclosure as part of the legitimation process.

As social media have become one of the most important instruments for public information, engagement and stakeholder relationship-building [25,26], depending on these objectives, the following three strategic approaches to communication in the CSR literature have developed: (1) a broadcasting [27] or information strategy [28], which does not contribute to relationship building but rather focusses on public information. Following a (2) stakeholder response [28] or reactive strategy, companies take opinions and tendencies from the milieu into account by passively reacting to user comments, questions or remarks [27]. When dialogue with stakeholders is fostered to carry out actions that result in some mutual benefit [29], a (3) stakeholder involvement [28] or engagement strategy is applied [30]. As two-sided or symmetric communication and relationship-building are core principles of public relations, they have also been highlighted further in studies on Twitter and CSR [27]. Co-creation is one example of an engagement strategy, with stakeholders being directly addressed via tagging to nudge them, either to reply or re-tweet, with each party contributing to the dissemination (by forwarding), and potentially to the construction (by modification) of the message [31].

Communicating good deeds and inviting stakeholders to engage in the conversations about a cause [32–35] can generate a stock of social resources referred to as social media capital (SMC) [36], which ultimately indicates communication success. Employee dialogue, for example, helps attract and retain talented employees [37] and strongly predicts employee engagement [38], motivation [39] and increased organizational commitment [40]. Some argue that retweets and likes are indicators of success that reflect not only to what extent the message resonates with online stakeholders [41] or impacts society [42,43] but

also increases customer loyalty [44,45]. SMC can thus expand to achieve organizational outcomes [33] such as favorable attitudes and better support-behaviors (e.g., purchase, seeking employment, investing in the company), build corporate image, strengthen stakeholder–company relationships by having a significant positive impact on customer identification and satisfaction [46], and enhance stakeholders' advocacy behaviors [47] and overall corporate reputation [48,49]. When stakeholders engage with each other, including in the exchange of relevant marketing information on a service or brand, to shape the most likely consumer behavior and attitudes towards products or the company itself, it is referred to as word-of-mouth (WOM) [42,50] As for Twitter, CSR topics are found to be a socialization agent that facilitate electronic word-of-mouth (eWOM). Adding the analysis of stakeholder sentiments and thus a directional (positive or negative) association with the eWOM activities [43,51] CSR topics in general are related to customers' positive WOM [52], which is thus another indicator of (digital communication strategy) success.

## 3. Material and Method

### 3.1. Data Collection

Five databases with complementary research areas and focus were selected for a literature review [53]: EconBiz (1) as the virtual subject library for the field of economics, ScienceDirect (2), covering topics of physical sciences and engineering, life sciences, health sciences, social sciences and humanities, (3) Taylor and Francis Online, adding a source for information science, mathematics and statistics, (4) MDPI (Multidisciplinary Digital Publishing Institute), focusing on open-access journal publication and (5) PubMed, for biomedical literature from MEDLINE. We decided to not search in Scopus and Web of Science as well, because there is a significant overlap, but to fill potential gaps via Google Scholar. We chose to limit publications to those that date back to not before 2013. We selected this period in view of the absolute number of monthly platform-users as well as the monthly growth-rate: in the first seven years (until 2013), Twitter was able to surpass 200 million monthly active-users worldwide for the first time. In the following eight, the platform was able to acquire around another 100 million users; however, the growth rate has since then leveled off. At the same time, Twitter Inc. became a public trade company in November 2013, which has transformed the way the platform is used in general. In addition, social media continues to evolve rapidly, so that studies prior to publication in 2013 could only examine the first few years (Twitter was founded in 2006), with an unrepresentative number of corporations using the platform specifically for CSR purposes at that time [54]. The keywords for our search strategy were identified independently by both authors as part of preliminary exploratory feasibility-searches. In an iterative process, keywords were identified and condensed to define the two subject areas as components of a compound search: (A) communication channel (keywords: Twitter with the synonym tweeting) and (B) communicated content (keywords: CSR with the synonyms Corporate Social Responsibility, Sustainability, and Corporate Citizenship) in agreement with each other.

After applying this search strategy in the five stated databases, resulting in 187 initial identified articles, as well as in an additional search in Google Scholar (adding eight publications), we first removed duplicates (45) as part of the identification process. We then systematically screened results according to the PRISMA 2020 (Preferred Reporting Items for Systematic Reviews and Meta-Analyses) flow diagram, as follows (Figure 1) [55,56]. Defined criteria for inclusion were: (i) findings had to be published as a journal article, review, case study or as conference proceedings, (ii) research must include companies/brands or their direct stakeholders, and (iii) papers using data from Twitter and other social networks were included if the analysis also included an individual consideration of each platform. Literature was excluded if one of the following applied throughout the screening process: (i) focus on Twitter as a company and its CSR activities, (ii) research with no clear Twitter focus, (iii) research discussing political topics or sustainability, with no clear CSR focus. Applying the defined criteria, we removed inaccessible studies (1) and inappropriate results

after abstract (94) and full-text (13) screening, leaving 42 relevant publications for analysis in our review.

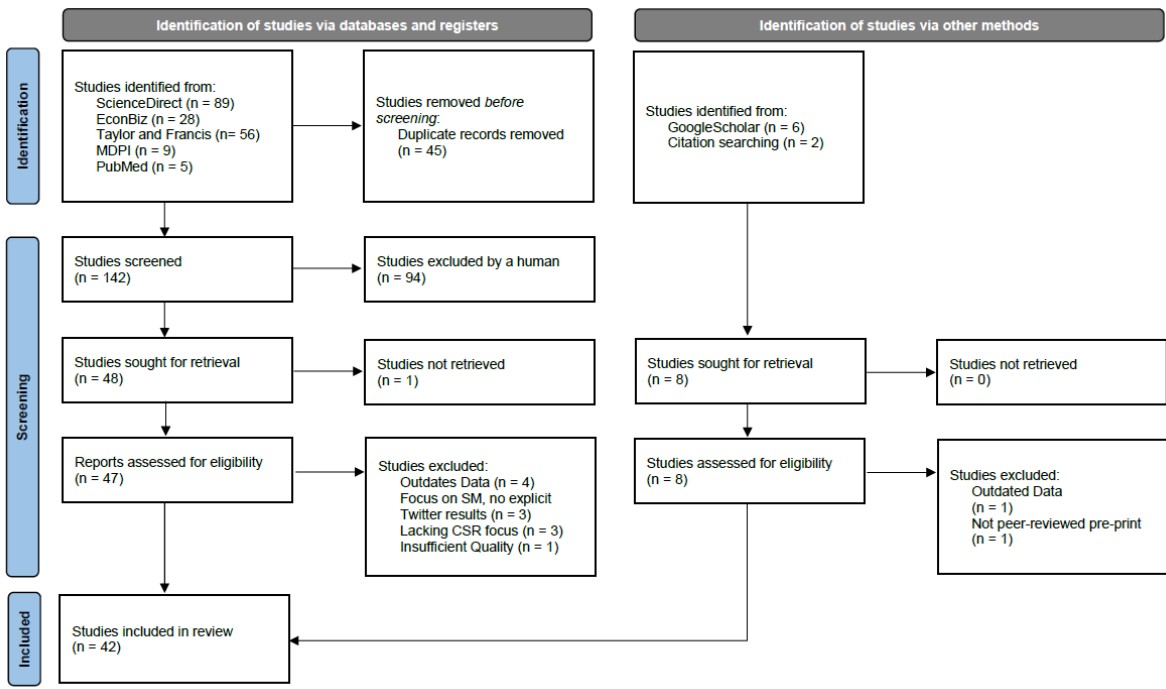

**Figure 1.** PRISMA Flow Diagram 2020 [56].

*3.2. Data Analysis*

The applied data analysis corresponds to successfully applied methodologies in the context of systematic reviews as well as CSR topics [57,58] and was chronologically organized as follows [53]: first, one of the authors read all of the articles and coded them, based on a coding manual. Codes for sample description included (i) publication year, (ii) target group, (iii) geographical focus, (iv) sample-selection criteria, and (v) sample size. For code generation regarding CSR topic clusters, we followed the sustainable development goal (SDG) wedding cake by Folke et al. (2016), which enabled us to cluster CSR topics using the sustainable triple bottom line [59] (environmental, economic, and social topics). For context: in 1992, the United Nations defined three pillars of sustainability at the Rio Conference, also referred to as sustainable triple bottom line [60]. It describes a balance of social, economic, and environmental goals that must be pursued simultaneously and reconciled, to ensure sustainable development [61]. For the first time, the importance of economic gains and societal benefits formally became the guiding principle of international policy. In 2015, United Nation (UN) member states committed to achieving 17 Sustainable Development Goals (SDGs), on a national level, which covered aspects of the sustainable triple bottom line, as part of the subsequent Agenda 2030 [62]. Although the triple bottom line seems to be gaining acceptance in the context of CSR in science, politics and practice, there were already scientific cluster-approaches and definitions for CSR topics before the conference in Rio, which had evolved over time and in some cases complement or stand alongside the triple bottom line:

In the context of manifesting a universally valid definition of corporate social responsibility, Archi B. Carroll developed the corporate social performance (CSP) approach as early as 1979, based on the 3-phase model described by Sethi in 1975, which classifies an adaptation of corporate action to social needs [63,64]. Later, Carroll (1991) identifies four areas of social responsibility for companies, and presents them within an explanatory perspective he has conceived by means of a fixed hierarchical order represented by a four-level pyramid. He describes the first two levels, which include economic and legal responsibility and which at the same time form the foundation of the pyramid, as fundamental

demands of society to ensure a company existence [65]. Level three outlines a company's ethical responsibility, equivalent to social expectations. Philanthropic responsibility, which encompasses an exclusively voluntary commitment that goes beyond the expectations of society, stands at the top of the pyramid. Carroll's three-domain model from 2003 is a refinement of the four-level pyramid, in which both the hierarchy and the philanthropic level as a whole are dropped, as part of a conceptual change. This decision is based on the acknowledgement that companies voluntarily engage in social activities for both economic and ethical motives, as well as a combination of both, and that philanthropy can therefore be subsumed as part of the task. A Venn diagram graphically represents the relationship between the remaining three dimensions of responsibility—economic, legal and ethical—by visualizing intersections, resulting in a total of seven possible categories of corporate social responsibility [66]. Thus, for CSR topics initially defined according to Schwartz and Carroll's (2003) three-dimensional model (economic, legal and ethical) or Caroll's CSR pyramid (1991) we relied on a topic classification framework that matches the 17 SDGs to the three-dimension-model [67], to then draw again on the CSR wedding-cake for clustering SDGs for the CSR triple-bottom-line.

For CSR strategy identification, we drew on the systematization proposed by Etter (2013) and explained earlier (broadcasting, reacting, engaging) [27]. The coding manual was refined gradually during the coding process, and comprised several stages for codes that needed interpretation (especially communication strategies). We ensured intercoder reliability via double coding by one of the authors and a third researcher who was not part of the data research and screening process. To assess the reliability of the coding scheme, we compared their coding for n = 8 studies (19% of our sample) by computing Krippendorff's Alpha [68]. Intercoder reliabilities lay between 0.86 and 0.95. Deviations were subject to discussion until the authors agreed on which coding was appropriate and consistent with the remaining coding.

## 4. Results and Discussion

### 4.1. Sample Descriptives

Examining our studies in terms of geographic focus, which includes both country-specific questions and a national delimitation of the study group (company headquarters or focus market), shows that 21 studies had a national and five an international focus, by including at least two countries in their sample selection. This was the case when either a comparison of two distinct cultural, political, or economic systems was of research interest, or a logical similarity of markets due to geographical proximity or international business strategies was given. At least one country from each of the five continents was explicitly considered. Specifically, a total of seventeen different geographic foci (countries) were studied, with five in Europe: Germany, Ireland, Italy, Spain, United Kingdom (UK); three in North America: United States of America (USA), Canada, Mexico; four in Asia: People's Republic of China, Japan, India, Turkey; three in South America: Republic of Peru, Chile, Colombia; one in the African continent: South Africa; and Australia. Figure 2 shows the different geographic foci (countries) identified, and adds the number of studies specifically related to that country, e.g., for the U.S., twelve studies and for Australia, one.

In addition, sixteen studies did not have an explicit geographic focus, but included a sample of either globally operating companies or user tweets within the entire platform, which were then further narrowed down, e.g., by their own systematization, including filtering by CSR-relevant hashtags. Thus, in these cases, CSR-relevant content in English was explicitly studied without subsequently describing the sample based on geographic characteristics, which is why we classified these studies as global.

The further analysis of included target groups included the assignment to an industry, applied sample-definition parameters, and sample size. Table 1 provides an overview structured by publication year. The following numbering refers to Table S1, which contains further sample information and is available as Supplementary Materials.

**Table 1.** Target group descriptives.

| Publication Year | No. | Consumer/Twitter user/Citizen | NPOs | CEOs | Brands | Firms/Corporations | Industry/Sector | Names Disclosed | Sample Selection Criteria | Sample Size (Twitter Accounts) | Sample Tweet Number (Tweets/Retweets/Replies) |
|---|---|---|---|---|---|---|---|---|---|---|---|
| 2022 | 15 | ✓ | | | | ✓ | | | CR Magazine' 100 best Cz | 71 | 22.951 |
| | 23 | ✓ | | | | ✓ | | ✓ | Diversity Inc Top 50 | 5 | 2.217 |
| | 42 | ✓ | | | | ✓ | Container Shipping | ✓ | not disclosed | 8 | 6.566 |
| 2021 | 1 | | | | | ✓ | | | FTSE 350 | not disclosed | 67.908 |
| | 2 | ✓ | | | | ✓ | | ✓ | Listed in Major Global Stock (15 SE listed) | 483 | 4.484 |
| | 7 | ✓ | ✓ | | | | Education | ✓ | individual systematic | 1 | not disclosed |
| | 10 | | | ✓ | | ✓ | | ✓ | DAX 30 | 36 | 154.770 |
| | 11 | ✓ | | | | ✓ | News Media | ✓ | individual systematic | 6 | 6.666 |
| | 12 | ✓ | | | | ✓ | Banking | ✓ | individual systematic | 41 | 2.816 |
| | 14 | ✓ | | | | ✓ | Consumer Brands | ✓ | individual systematic | 3.093 | 44.432 |
| | 16 | ✓ ** | | | | | Alcohol | n.a. | n.a. | 175 ** | n.a. |
| | 26 | ✓ ** | | | | | | n.a. | n.a. | 219 ** | n.a. |
| | 27 | ✓ | | | | ✓ | | | AIDA | 417 | 917.864 |
| | 29 | ✓ | | | | | Mining | n.a. | n.a. | not disclosed | 2.000.000 |
| | 33 | ✓ | | | ✓ * | | | ✓ | Fortune 200 | 42 | 163.402 |
| 2020 | 21 | ✓ ** | | | | | Alcohol | n.a. | n.a. | 839 ** | n.a. |
| | 25 | ✓ | | | ✓ | | | ✓ | individual systematic | 8 | 428.000 |
| | 31 | ✓ | | | | ✓ | | ✓ | Blue Chip Companies (EuroStock 50) | 50 | 127.811 |
| | 34 | ✓ | | | | ✓ | | | Fortune 150 | 41 | 1.079 |
| 2019 | 13 | | | ✓ | | ✓ | | | Fortune 200, Hootsuite | 93 | 194.644 |
| | 17 | | | | | ✓ | | ✓ | National CSR Ranking by IIM Udaipur (India) | 34 | 4.091 |
| | 20 | ✓ ** | | | | | Alcohol | n.a. | n.a. | 177 ** | n.a. |
| | 22 | | ✓ | | | | Alcohol | ✓ | not disclosed | 6 | 1.805 |
| | 28 | ✓ | | | | | | n.a. | n.a. | 223.476 | 414.926 |
| | 32 | ✓ | | | ✓ * | | | ✓ | Fortune 500 | 38 | 1.125 |
| | 38 | ✓ | | | | ✓ | Airline | ✓ | individual systematic | 6 | not disclosed |
| | 39 | | | | | ✓ | Tobacco | ✓ | individual systematic | 4 | 3.301 |
| | 40 | ✓ | ✓ | | | | | | Council on Foundation List on Website | 198 | not disclosed |
| | 41 | ✓ | | | | ✓ | | | Forbes Ranking of 2000 Largest Corporations | 30 + 54 FB accounts | 2.672 |
| 2018 | 4 | ✓ | | | | | | n.a. | n.a. | not disclosed | 178.908 |
| | 6 | | ✓ | | | ✓ | | | Cone Non-Profit Power Brand 100 | 65 | 5.859 |
| | 30 | | | | ✓ | ✓ | Consumer Apparel | ✓ | Newsweek's Greenest Companies Rankings | 11 | 187.177 |
| | 35 | ✓ | | | | ✓ | Banking | ✓ | individual systematic | 2 | 2.719 |
| | 36 | | | ✓ | | ✓ | | | MERCO Ranking | 93 | 1.657 |

**Table 1.** *Cont.*

| Publication Year | No. | Stakeholder | | | | Companies | Industry/Sector | Names Disclosed | Sample Selection Criteria | Sample Size | Sample Tweet Number |
|---|---|---|---|---|---|---|---|---|---|---|---|
| | 3 | | | ✓ | | ✓ | | ✓ | individual systematic | 63 | 32.641 |
| 2017 | 18 | ✓ ** | | | | | | n.a. | n.a. | 507 ** | n.a. |
| | 37 | ✓ ** | | | | | | n.a. | n.a. | 253 ** | n.a. |
| 2016 | 5 | ✓ | | | | ✓ | | ✓ | IBEX 35 | 20 | 5.106 + 416 FB postings |
| 2014 | 9 | ✓ | | | | ✓ | | ✓ | CR Magazine' 100 best Cz | 30 | 41.864 |
| | 8 | ✓ | | | | ✓ | | | CR Magazine' 100 best Cz | 30 | 41.864 |
| 2013 | 19 | ✓ | | | | ✓ | | | Fortune 500 | 222 | not disclosed |
| | 24 | ✓ | | | | ✓ | | | IBEX 35 | 35 | 5.352 |
| Total | 42 | 32 | 5 | 3 | 4 | 27 | | 21 | | 228.682 | 5.051.509 |

* specific corporate CSR accounts, ** experiment participants. ✓ included.

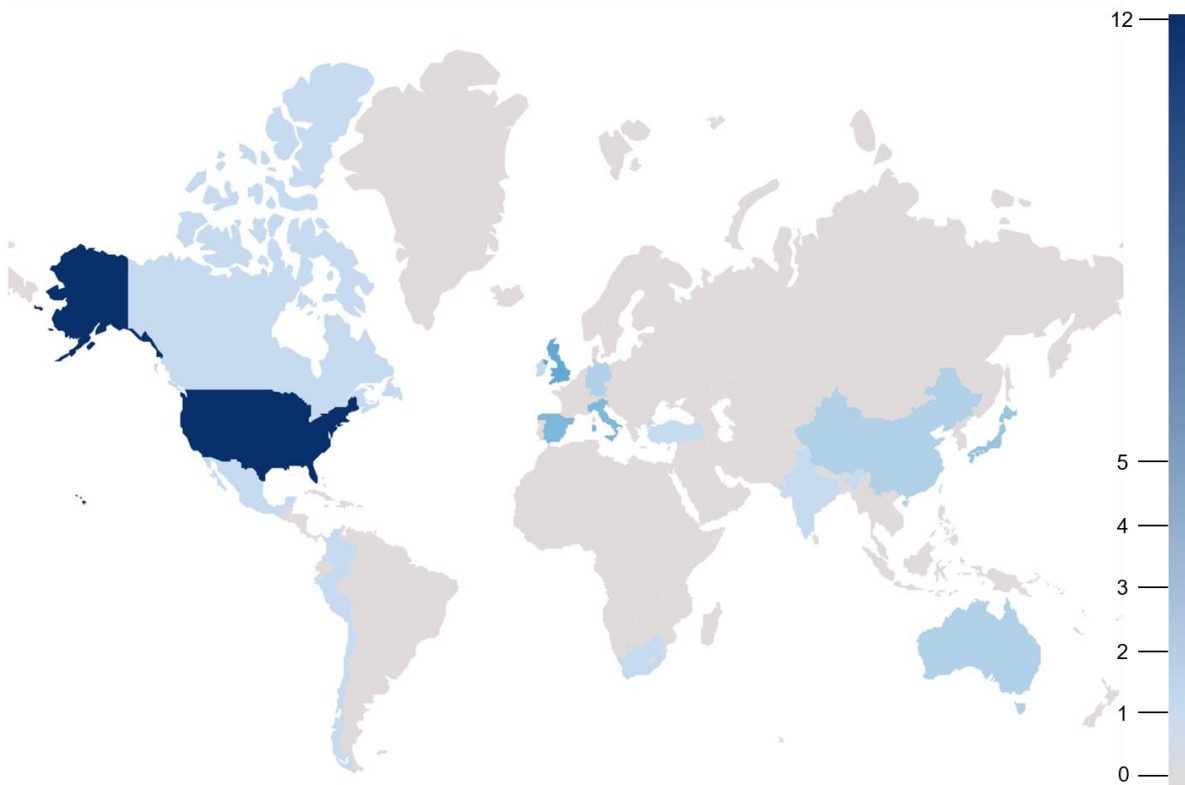

**Figure 2.** Geomap of national study focus.

Approximately 31% of the reviewed studies focus on one target group, of which three focus on companies and nine on stakeholders, with six of these being experiments (quantitative surveys) involving participants. Thirty studies thus cover two target groups. Overall, and regardless of whether a single or two target groups are focused on, stakeholders represent the target group in 38 of the 42 studies, with 32 including the more general and thus also largest subgroup of consumers, Twitter users and citizens with no further specification. Five studies focus on non-profit organizations (NPOs) and three on chief executive officers (CEOs). Thirty studies include companies as the target group, twenty-seven focus on corporations or firms, and four on brands, including two studies that focus on the company's respective specific-CSR Twitter accounts. From a stakeholder-theory perspective, it is evident, that studies focus on external as well as internal stakeholders, neglecting employees or employers. Duarte et al. (2014) found that the perceived level of engagement in socially responsible practices contributes to triggering the process that leads individuals to evaluate an organization as a good place to work [69]. Thus, CSR can be a source of competitive advantage regarding the recruitment of new employees, and we conclude that the explicit consideration of employees in studies on CSR activities on Twitter is an underrepresented approach in the research, which needs to be filled with future studies (including the means of employee needs identification).

Altogether, fourteen studies focus on nine specific industries: alcohol industry (4) consumer brands/apparel (2) banking industry (2) education sector (1) container shipping industry (1) mining sector (1) airline industry (1) news media (1) and tobacco industry (1). Industries linked to below-average working conditions regarding health and safety and environmental issues on emissions are focused upon. With the mining industry in particular being one of the first to start rethinking their actions for workers' health in the 1950s and 60s by investing in health facilities in the US, CSR-related actions have a long history, and are still waiting to be resolved [70]. These findings are in line with the legitimation theory suggesting that companies operating within these kind of industries legitimate their actions and business by increasing the amount of environmental disclosures

in their annual reports or on social media, which is then not infrequently discussed as greenwashing [24].

For 21 out of 33 studies in our sample (excluding studies focusing on twitter *users* alone), the company's name (profit or non-profit) or their twitter-account name is disclosed, which contributes to greater transparency and possibility of study replication for longitudinal approaches.

For the same 33 studies that contain the actual study-group names, it would thus have been possible to provide information on the applied criteria in the sample composition, with only two studies not using this option and failing to specify sample-selection criteria. Nine studies provide their own systematization logic in the sample definition. Fourteen studies determine their sample based on a grouping/listing outside of Twitter that focuses on an economic ranking (national or global stock-market listing (6), Fortune 150 to 500 (5), Forbes 2000, Cone Non-Profit Power Brand 100, and AIDA (national Italian corporate database) (1 each)). Finally, eight studies choose a sustainable-related ranking for sample selection (CR magazines' 100 best Corporate Citizen (Cz) (3), Diversity Inc. Top 100, Newsweek's Greenest Companies Rankings, Council on Foundation List on Website, MERCO (Monitor Empresarial de Reputación Corporativa) ranking (for South America), National cCrporate CSR Ranking by IIM Udaipur (India) (1 each)). The decision for such a deductive approach can bring the advantage of having a sample already confirmed as acting sustainably, and thus increase the chance of finding appropriate content on Twitter.

In addition, 92% (33 of 36 studies: the six experiments also report the number of participants) provide information on the number of Twitter accounts included in the final sample (ranging from 1 to 223.476; mean: 6.932). A total of 89% (32 of 36) provide information on the number of tweets, retweets, and replies included in the analysis, ranging from 1.070 to 2 million (mean: 158.646). The range of sample size makes the associated method-variability evident. Since the chosen quantitative and quantitative methods, as well as the associated use of machine learning approaches or the combination of these produce extensive results, this area forms a further approach for future research.

Looking at the publications over time, we see an increasing trend since 2013, illustrated in absolute numbers, with a peak in 2021 with a total of twelve publications, in Figure 3 section A. In addition, research topics covered two main areas: the identification and analysis of CSR-relevant content (in 22 studies: 52%), and communication strategies (28 studies: 67%).

Research on CSR-related content only came to the forefront towards the end of the 2020s, with a continuous increase in interest since then (Figure 3, section B). A likewise continuous increase can be observed in studying communication strategies, which, however, already took off approximately five years earlier. This may be related to an initial interest in understanding social media communication in general, with CSR being an applicable context approach, as content was easily identifiable by unique tags and Twitter had been known as a corporate communication channel since the platform's early days. However, with the introduction of international laws and guidelines on sustainability reporting, the actual content has gained interest for developing the implications for reputation-enhancing actions. At the same time, increasing the low-threshold possibilities for qualitatively analyzing substantial amounts of data automatically, could explain this trend. One conclusion could therefore be that the benefits of social media in the context of legitimation were not initially recognized by companies, and are now increasingly developing as a hygiene factor.

We were able to differentiate the two identified research strands (CSR content and communication strategies) into additional subcategories (Diagram C in Figure 3). Studies that investigate CSR-relevant content, for example, can be further subdivided into those that investigate whether CSR-related content was the subject of Twitter communication at all, following a CSR-topic analysis (63%) (CSR disclosure). In cases where the Twitter activities on CSR topics were already well known, the analysis then focused on the specific content only (23%). Results of both subcategories are presented in Section 4.2.1. Finally, influences on topic choice and CSR disclosure in general, e.g., corporate characteristics are covered in

Section 4.2.2. Research on CSR communication strategies revolves around communication direction, whether this is 1- or 2-sided (39%). The impacts of communication-strategy choice or corporate characteristics on social media capital (SMC) (32%) is another field of interest, as marketers and managers are continuing to look for ways to measure the return on investment as regards social media activities. Co-created communication as one specific 2-sided communication strategy (12%), and how corporate characteristics (e.g., diversity of the board) can influence communication-strategy choice (11%) and impacts on word-of-mouth (7%) are further subcategories found, addressed in Section 4.3.

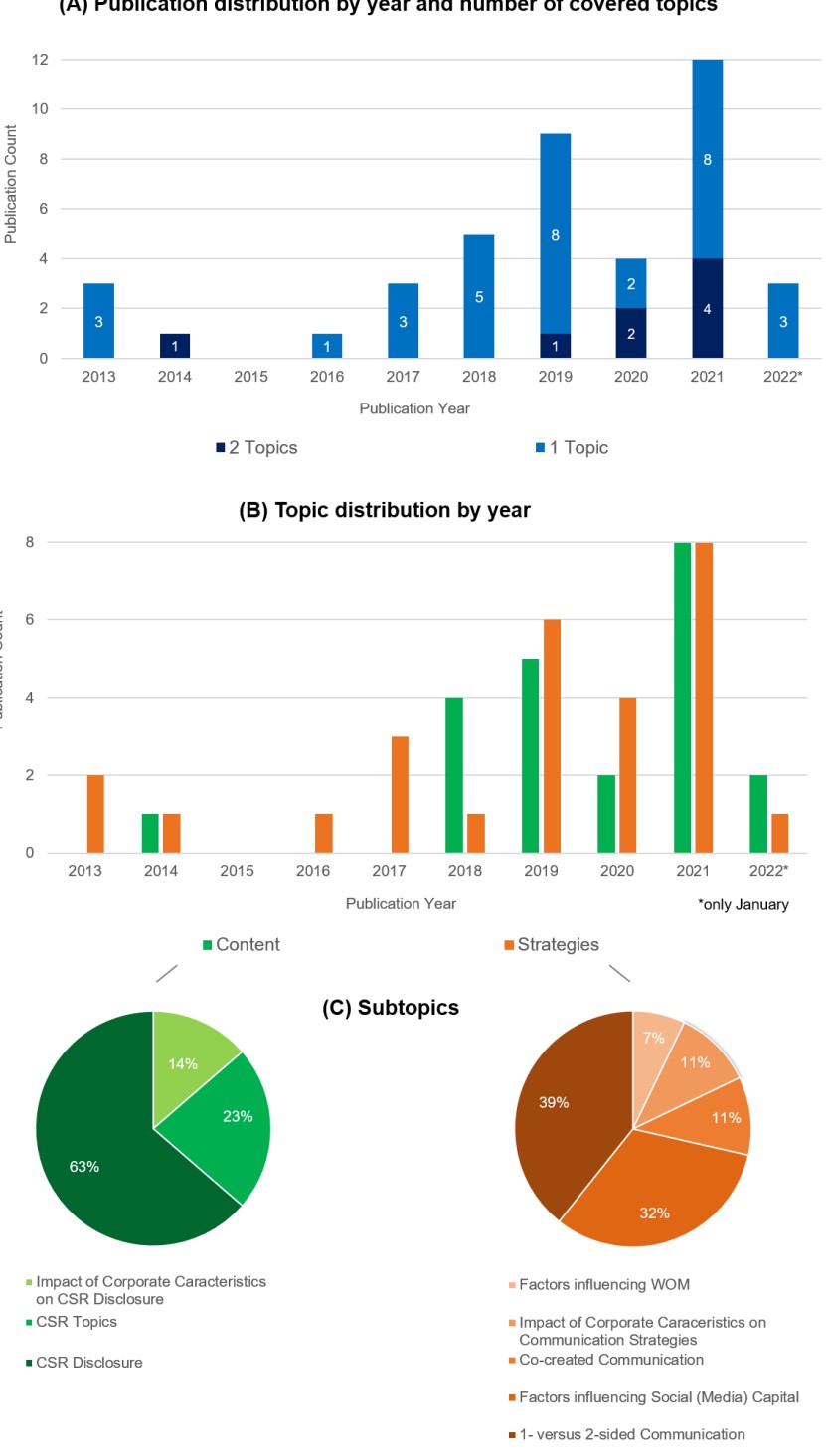

**Figure 3.** Time lapse of research topics.

*4.2. CSR Topics*

Eight studies chose a deductive approach with the assignment of identified CSR topics to a predefined-category system: Carroll's CSR pyramid, the three-dimensional model by Carroll and Schwartz, SDGs, the ESG approach (environmental, social, and corporate governance: an umbrella term for evaluating and identifying sustainable investments) and the Global Reporting Initiative (GRI) standards. For context: to support firms in high-quality and reliable sustainability-reporting, developed standards try continuously to draw on international legal frameworks such as the SDGs to enable transparent, comparable and comprehensive reports [71], e.g., as an analytical measurement of negative and positive impacts within the value chain [72,73]. One relevant example for a CSR-reporting standard provider is the Global Reporting Initiative (GRI). Founded in 1997, it has served as an internationally independent organization up to today, whose self-proclaimed goal is to enable a comparable and standardized presentation of the environmental, social, and economic activities of large corporations, small and medium-sized enterprises (SMEs), and other organizations and governments [74]. Due to early cooperation with the UN Global Compact, which recommends the application of the GRI standards to its members, it has established itself as suitable for the preparation of sustainability reports globally. The GRI standards thus support report preparation based on the UN SDGs.

By analyzing our sample, we observed that drawing on an established CSR-categorization system can facilitate cross-industry and cross-country comparisons on the one hand. On the other hand, our review results show that analyses following a deductive approach often remained at the superordinate level of category designations. Studies adopting an inductive approach yielded more in-depth content concerning specific CSR-related measures and campaigns which can contribute significantly to the understanding of communication strategies on Twitter. Similarly, research that focused solely on a sub-area of CSR, such as diversity or a specific industry, also provided more substantive results than those looking at the entire spectrum of CSR communication. Future research designs can build on that finding when choosing a methodological approach and defining sampling and topic-cluster approaches.

As explained in our Method section, to cluster communicated CSR topics on Twitter, we drew on the CSR triple-bottom-line, within which the 17 SDGs can be found. For the visualization, we decided to assign the identified topics to the individual levels of the CSR wedding cake (representing the triple-bottom-line and SDGs). Figure 4 thus shows the respective studies that contain one or more content areas (left); (again, numbers are a reference to Table S1 as supplementary material with further information on the individual studies), as well as a quantitative overview in individual histograms of further identified top subtopic areas (right).

4.2.1. Social, Environmental and Economic CSR Topics

Social aspects are not only identified as a topic communicated on in most studies (Figure 4), but are also predominantly classified as focal topics within each study, as they, for example, generate the most retweets [75]. The internal stakeholder group of employees takes on a special status, as 52% of the studies identified communicated topics of (i) employee relations such as working conditions, career and education, social security, remote-work opportunities, employee health, labor practices and safety. The discussion of (ii) diversity and inclusion, as well as measures to promote the equality of minorities or women, is another key topic, as we learn about specific campaigns to combat violence against women, for example, as well as special programs to promote women in leadership positions. Companies emphasizing the benefits (increased creativity and problem-solving) of diverse teams in the field of innovation, point out their focus on minority hiring as well as their own commitment to breaking social stigma against, for example, transgender people. Issues related to (iii) community, locality or solidarity are the third key subtopic, including the promotion of charitable actions by donating goods to local communities. Another 29% covered the area of (iv) philanthropy, often related to employee volunteering

or monetary- and product-donations to social causes. (v) Human rights are identified in five studies, but play a subordinate role in each case, as they only account for around 5% of the total CSR discussion.

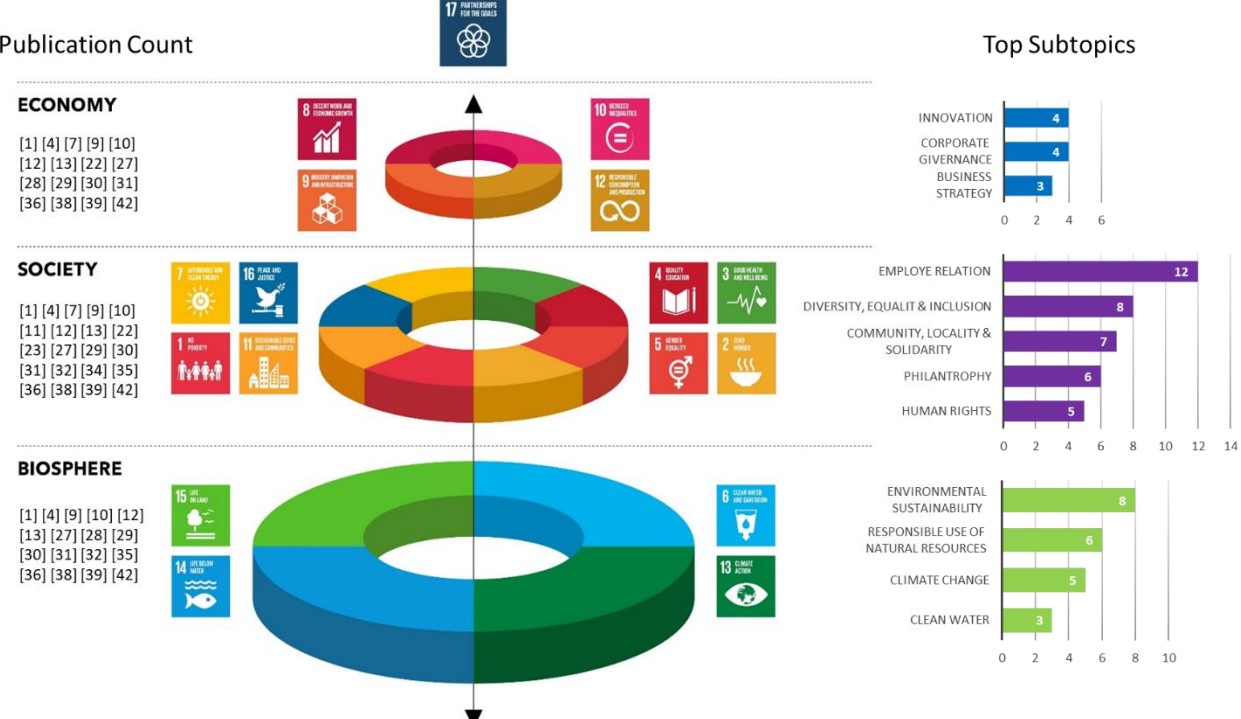

**Figure 4.** Topic and subtopics distribution along the CSR triple-bottom-line employing the SDG wedding cake [59] (Credit: Azote for Stockholm Resilience Centre, Stockholm University CC BY-ND 3.0). (Table S1: Sample).

We found that environmental topics are focused on less, which could be because environmental sustainability is becoming more of a hygiene factor rather than being a differentiator. In the 47% covering ecological CSR issues, topics cover (i) environmental sustainability. Six studies identify the topic of (ii) responsible use of natural resources by, e.g., setting targets of lower paper-consumption. (iii) Climate change is an identified topic in five studies, with reports on environmental projects, green initiatives or measures taken such as reduced business-travel. (iv) Marine pollution and thus the issue of clean water or life below water can be identified in three studies.

In general, economic topics related to CSR are the least talked about on Twitter when looking at the results of each topic cluster in our sample. We identified economic topics in seventeen studies, with (i) corporate governance being the broadest topic and the most present one, even though it has hardly any significance within the respective study, as it sometimes only accounts for around 2% of CSR tweets. Likewise, four studies found that the topic of (ii) innovation, often in the context of technology and digitization, was the most popular in the field of economics, with a higher intensity within individual studies, while (iii) business strategies in the context of CSR could be identified in three studies. This topic also takes on greater importance in each case, based on assigned tweet frequencies

It remains to be noted that, based on the topic frequencies, within the topic clusters along the CSR triple-bottom-line, those that can generate a direct concern of stakeholders on Twitter also referred to as human interest, stood out [76]. Topics that have no implicit direct benefit for stakeholders on Twitter, such as human rights, governance, or clean water, are identified less frequently. It is clear that sector-specific studies go into greater detail thematically and focus on different key topics depending on the sector, more precisely those that are most relevant within the industry.

4.2.2. Reasons for and Shift of CSR-Topic Disclosure

Looking at an overall trend determined by post-quantity, Amine et al. (2021) point out that over the course of time (from 2008 to 2016), companies in the UK have been adopting Twitter more frequently to disclose general CSR information, with an increase in CSR-related posts from under 1, to over 50 % [77]. Esposito et al. (2021) make the same observation in a case study from the University of Salerno in Italy between 2015 and 2021 [78]. Further, it is Chinese companies that have a strong preference for incorporating CSR subjects in Twitter communications, which is not surprising, as China is one of the leading export nations and largest locations of international production facilities, meaning that Chinese companies in particular have also discovered CSR for strategic communication with regard to requirements for supply- and service-chains [79]. Narrowing down the sample to the Euro Stock 50 companies, Twitter became an increasingly used communication channel since 2009, starting with 34% having a Twitter account and reaching 96% in 2016 [54]. The number of CSR-related tweets also increased exponentially; in 2009 only 12% of the 17 companies with official accounts were engaged in CSR communication, while the number was already 88% in 2016 [54]. However, the research-design chosen did not allow conclusions to be drawn on company improvements or specific projects regarding environmental and social impact or their implementation over time. It also did not examine a possible trend in, or change of popular CSR topics, over time [54]. Interestingly Okazaki et al. (2020) found that the majority of brands' CSR-labeled tweets' content are not even relevant to CSR [80], which underlines the importance of CSR topic-definition in regard to stakeholder interest and also legal frameworks. When looking at whether CSR is a topic communicated by CEOs of large companies on Twitter, results show that they, similar to corporate Twitter accounts, also do not employ the platform for CSR communication in general [67].

A longitudinal study of global consumer-apparel-companies' sustainable-environmental communication between 2010 and 2014 explains that corporations which are already ranked as good corporate citizens adopted CSR Twitter communication earlier than those with a bad reputation, and post more frequently [81]. Further, sustainability performance and CSR ranking have a positive influence on the quantity and quality of information shared on Twitter, as high-ranked companies will be more likely than low-ranked firms to demonstrate transparent external CSR-communication, as evidenced by participation in the voluntary GRI [81]. Suarez-Rico et al. (2018) found that firms operating in environmentally-sensitive industries present higher levels of CSR disclosure on Twitter than those in other sectors [82]. Further, there is a positive relationship between a firm operating in a sensitive industry and its level of CSR disclosure on Twitter, and an inverse relationship between the latter variable and the tenure of the CEO [82]. A mining-industry-specific analysis supports this finding by pointing out that CSR-topic discussion among all Twitter users is increasingly growing, especially in developing countries and in countries with a bad reputation for environmental and health (mining) conditions [83]. The authors conclude that CSR appears to be an evolving construct in business and society and its dimensions and trends change over time, although they don't outline the topic changes [83]. The absolute increase in CSR-related Twitter communication can also be observed in the narrowing down of the research focus from a general CSR-perspective to a specific subtopic and even an additional geographical focus. Thus, socio-political-positioning posts of German stock-exchange companies and their CEOs also continuously increased between 2014 and 2019 [84]. While in 2014 an average of 6.62 posts per month with socio-political positioning accounted for only 0.00282% of all (analyzed) tweets, 76% of the socio-political-positioning posts originated in 2018 and 2019. These studies underline the rising popularity of Twitter, indicating its growing importance as a CSR-disclosure platform in general [77]. In this context, Amin et al. (2021) discovered that the presence of women on company boards, especially non-executive women, is positively associated with the extent of CSR disclosure on Twitter [77]. At the same time, larger firms are also found to disclose more

CSR information on Twitter compared to smaller ones. indicating that firm size plays an important role in determining a company's level of CSR disclosure on social media [77].

Looking at actual changes of topic popularity between 2013 and 2016 among all Twitter users, Chae and Park (2018) showed that CSR-topic proportions have been, in fact, fluctuating over the years [18]. They outline a growth in popularity regarding social topics such as employee engagement and community charity. Building topic-clusters, analysis indicates that the community-oriented topics of the philanthropic cluster has a strong positive trend [18]. As gender shift is an emerging megatrend and sustainability is increasingly becoming a hygiene factor, at the same time the ethical cluster (a discussion of ethics in strategy and social-business contexts) is steady, and the environmental cluster is even declining over time [18]. As diversity activism increased in Western societies, US corporations also changed their approach to diversity notably, over time, when in 2017 they stressed the benefits of diversity in their operations via a business-centered diversity-communication approach, and switched to socially responsible corporate-diversity-communication by 2021 [13]. When drawing conclusions from CSR topic investigations on Twitter, and taking a greater period of time into perspective for analysis, it is crucial to understand that topic relevancy may change over time, and neglecting this development can lead to inappropriate strategic derivations for companies. Future studies could address the question of which time-horizon yields the best results, relative to model quality, depending on the availability of existing datasets (specifically for automated qualitative-analysis methods).

*4.3. CSR Communication Strategies*

When exploring digital CSR-related communication on Twitter, research primarily focusses on the identified strategies employed, in regard to a 1- or 2-sided approach. Thus, we will summarize findings regarding strategy choice at first, and will then focus on detected success-factors. As the analysis is complex and the approaches often employ machine learning approaches, which are the most heterogeneous, the presentation of our results thus does not focus on the methodological approaches, but rather on the empirical results affecting communication-strategy success. Future studies could address this and investigate which mythological approaches are employed, and shed light on the limitations and advantages of each method, systematically.

4.3.1. Communication-Strategy Choice

**Broadcasting**

The majority of companies tend to use social media similarly to other mass communication channels—by mainly distributing information in a one-way or asymmetric communication-approach, and not engaging the stakeholder in their tweets [14,27,29,54,80,83,85,86]. Although Twitter is a social network, the full potential of its interactivity is thus not being used [86]. By only disseminating CSR-related information, companies do not foster communication commitment, and show no interest in establishing stakeholder relationships through the identification of, and focus on, shared goals and common interests [27]. Mamic et al. (2013) even put a price on the possible costs for businesses due to the absence of an active listening and thus dialogic communication-strategy on social media, which may even reach up to one million euros daily [86]. Cortado et al. (2016) therefore highlight the necessity to change the way companies communicate their CSR issues, by shifting to two-way communication, as has already been the case in other enterprise relations [29].

**Reacting**

At the same time, Kaul et al. (2019) observe a slight shift from broadcasting to the adoption of reacting- and engaging-strategies [87]. Rodríguez et al. (2020) detected that 61% of large European companies still employ a one-way information strategy, but 39% employ a two-way response strategy, and none employ a two-way collaborative strategy [54]. The banking and finance sector especially, appear to be most prone to online CSR-communication-strategy adaption, which has also shifted to being a highly reactive one [87]. This may be due to societal and regulatory pressure to report on ESG (en-

vironmental, social and corporate governance) criteria to support transparency within a sustainable investment portfolio for potential shareholders, and to counter negative feedback. When questioning whether a stakeholder message has certain characteristics which make corporate Twitter accounts feel the necessity to interact, Saxton et al. (2021) found with data from 2014 that corporate-Twitter-account responsiveness is positively associated with stakeholder urgency in terms of both the originality of a stakeholder message and the expression of positive sentiment [88]. By at least responding to questions or remarks, companies tap in the potential to establish stakeholder relationships over time, and foster beneficial outcomes, such as trust, involvement, and commitment, on both sides [27]. Using a digital questionnaire, Kollat et al. (2017) highlight the fact that asymmetric communication efforts lead to higher trust in the company, compared with symmetric dialog-centered communication [14], as consumers do not appreciate the feeling of excessive company-engagement, especially when they detect self-promotion [14]. Results are limited, as the amount of online dialogue is highly dependent on the topic of interest. With a fictitious company, and participants evaluating use cases they in reality might not even engage with, bias can result regarding attitudes towards an online communication strategy.

**Engaging**

Although a shift in strategy choices in observed, it is still only a minority of companies on Twitter that approach their network proactively via, for example, tags, and nurture a real dialogue to open the possibilities for online relationship-management [27]. When looking at brands, for example, dialogues among consumers are present, but the brands themselves are rarely part of the conversation [80]. Looking at the implementation of co-created communication, there is only limited evidence [31,89]. We only find that social CEOs do activate their community and are thus able to leverage their reputation [67]. Companies are thus not tapping into the potential for co-creation, by not adding mentions of individual consumers nor audience-specific and relevant messages, which are inherent to social media [80].

An accelerator for a dialogic strategy, e.g., in the educational sector, was the rise of COVID-19, which led to the collapse of traditional, formal, one-sided communication and thus has imposed the need to converse with stakeholders on social topics. Social media became a major source of digital socialization, as in-person meetings were regulated, due to health safety measures [78]. Companies were thus pressured by a major shift in society and the acceleration of digitalization in every persona and in business life, adapting and overcoming former gaps regarding digital communication and interaction. Even after the pandemic, the use of social media for exchanging ideas, especially on social causes, is not expected to decline, as companies now more than ever increase their online communication by introducing to the workforce especially digital employer and consumer communication, which has come to stay and which will last.

**Influencing factors**

Lee e al. (2013) found that a high CSR-rating leaves companies in a favorable position to absorb the risks associated with the use of new media, and enables early adoption. The social support facilitates companies in establishing a greater online community in regards to follower count in a shorter period of time [90]. Baboukardos et al. (2021), for example, point out that companies with better social performance are also more likely to engage in, and hence communicate, stakeholder-oriented actions for the COVID-19 pandemic [91]. More precisely, this can be determined as corporate social advocacy (CSA), with companies responding to events or developments that are not self-initiated and affecting the company only indirectly if at all, as a transparent form of political engagement [92]. Adding findings by Froehlich and Knobloch, the larger the company, the more likely it is to engage in CSA communication especially, while companies with a B2B business model are less likely to adapt this kind of communication to Twitter [84]. The reasons are the greater public and legal pressure to face up to social responsibility in general, the greater political influence that comes with increasing company size, and the greater cognitive legitimacy that comes

with less vulnerability [64,84,93]. To provide the means for possible criticism regarding the CSR topics communicated, Etter (2013) found, that specialized CSR-company accounts react more efficiently and appropriately than general-company Twitter accounts through the allocation of staff- and time-resources [30]. CSR accounts thus do not only disseminate more CSR information, but also have a significantly higher level of interactivity [30].

4.3.2. Success Factors of CSR Communication Strategies

Communication on corporate giving such as philanthropic donation, especially with video content, increases social-media-engagement behavior through the number of retweets, comments and likes [78,83]. Moreover, socially responsible firms are able to harvest proactive stakeholders' participation, also referred to as user-driven communication or WOM, without investing more resources (firm-driven communication) [90]. In addition, by driving consumer-engagement behavior, CSR campaigns on Twitter have an influence on enduring audience, causing engagement behavior over a longer period of time [94].

Equivalent to our findings regarding CSR topic-choices and shifts over time, it is again special CSR-related events that trigger communication and engagement with the stakeholder on Twitter. Thus, specific periodically recurring events as well as crises are important and strategically necessary as implanted pillars in Twitter CSR-communication-strategies [84]. Besides the benefits of tapping into existing social-movement discussions, increasing engagement can be generated if communicated CSR topics are made explicit by the use of hashtags [95], although non-CSR posts with hashtags related to the triple-bottom-line are retweeted and liked less, since users may feel misled [75]. Patuelli et al. (2021) [75] find that stakeholder engagement measured by retweets and likes is in general not homogeneous, and varies depending on the communities, with the social dimension again scoring a higher number of retweets compared with the other CSR dimensions [75].

Overton et al. (2021) found that individuals inferred more value-driven motives from CSR messages that are directly connected to their actions compared with CSA messages, which ultimately creates a more positive attitude toward changes in the company and WOM intentions [96]. Since the findings were experiment-based, they will need to be backed up with an analysis of real Twitter data [96]. In that context, O'Brien et al. (2018), for example, also found that customers prefer firms addressing social issues which are aligned with their core purpose, and are then willing to be engaged by addressing them [44]. Interestingly, a company tweeting about converting part of its original production to, for example, cover the needs arising from the pandemic and to support for workers, were most appreciated by the Twitter community, leading to positive sentiment and WOM [97].

Further, for industries that are the focus of public attention due to their intrinsically environmentally harmful products and services such as aviation, digital CSR engagement on Twitter is a driver of positive WOM, which aligns with the concept, that customers are concerned about issues other than service quality and value for money [98]. Results are in line with Markovic et al. (2021), who demonstrate the positive affect of CSR topics on customers' positive WOM through brand authenticity [52]. Environmental and social CSR engagement especially, act as safeguards in reducing online negative WOM about the company, while economic CSR engagement has only a small effect [52,78]. Those companies following long-term consistency in approaching, for example, diversity topics on Twitter, receive more positive sentiments and higher engagement than companies that make swift changes in the aftermath of increased pressure from activist groups [13]. Additionally, communication on diversity topics that go beyond a company's own CSR objectives again increased positive user-responses [13].

Pons et al. (2021) found that environmental tweets with the word "environment" convey a positive sentiment, while tweets with the word "climate" mainly elicit a negative mood, since Twitter users associate climate change with the increase in severity and frequency of certain environmental disasters, and tweets containing the word "environment" mainly refer to environmental and sustainable practices, which are seen as valuable to fight against climate change and to improve the environmental behavior of companies [83].

Findings are in line with studies on framing effects, where cognitive bias influences peoples' decisions on options based on whether the options are presented with positive or negative connotations and attitudes towards a subject [99,100].

In addition it is the overall Twitter follower-count that shows a positive effect on user sentiment [101,102], with men especially being more influenced than women [103]. As social proof, the positive effect of a large network, among other things, in increasing credibility when it comes to purchasing decisions, is already known from other industries [104,105]. For women in turn, the sentiment towards a company's CSR content was positively influenced by the organization type being non-profit rather than profit [103]. Effective acquisition of social capital for non-profit organizations by CSR communication on Twitter is content- and connection-based [106]. Thus, it appears to rely less on the quantity of organizations' stakeholder engagement than on the diversity of that engagement, both in terms of stakeholder connections and complexity of message elements [106]. Adding to these experimental findings, a machine learning approach with real Twitter data confirms that a company with more direct connections with others is more likely to have an opportunity to obtain public support regarding positive user-sentiment [32]. Thus, network size is crucial to accessing social resources and to mediating the relationship between corporate retweets/response and stakeholder support [32].

When looking at an experiment comparing CSR message-content in terms of product-related messages designed especially for the stakeholder group of customers, framing information within a CSR context has a positive impact on consumers' purchase intention, rather than just highlighting product benefits and attributes [49]. At the same time, adding an ethical aspect to product- or community-relations-messages did not leverage attitudes towards the brand, the stakeholder engagement or even WOM intentions. Thus, Uzunoğlu et al. (2017) found only limited impact on consumer outcomes of CSR messages within both product and community relations [49]. Interestingly, also in this case, the experimentally-based findings contradict somehow the empirical findings employing Twitter data directly; therefore, we question whether experiments are the right choice of study design when investigating questions regarding stakeholder engagement or sentiment, since mutual interest and network size and positioning play key roles when measuring social media KPIs.

## 5. Implications, Limitations, and Conclusions

### 5.1. Implications

Our review results contribute to a better understanding of CSR-related communication strategies on Twitter by highlighting the content to prioritize, as well as beneficial techniques and tactics by summarizing, discussing and classifying current results in the context of communicated CSR topics and specific communication strategies on Twitter in different markets and across different nations. The review systematically captured the platform's past activities and strategies up to 2022, synthesizing practical information that can guide Twitter usage decision-making and be used for research to serve as the basis for future comparative longitudinal studies of changes in usage due to the Twitter acquisition by Elon Musk in the fall of 2022, which could probably lead to a change in platform use. The following is a summary of the implications for marketers from the review results discussed.

First, our review results show that when opting for CSR-Twitter communication, it is an all-or-nothing approach. To reach the right target group that is interested in CSR topics, study results show that it is recommended to establish a company account that clearly revolves around sustainability issues. In this way, complaints about products or services, but also other marketing activities, can be continued on the actual corporate account, thus minimizing the risk that CSR issues are perceived as a pure marketing-measure by online stakeholders. In addition, a second account can already enable co-created content by linking, sharing, and referring to topics among the different company accounts. It is then crucial to grow a community fast, as overall follower-count positively influences men's attitudes especially, towards the company.

Second, community connections need to be as diverse as possible, e.g., with employees, business partner or NGOs. To establish a strong connection with stakeholders, it is essential to not only initiate but to find the dialogue on the company's own CSR account, by directly addressing relevant stakeholders via tags or mentions, but also on stakeholder accounts, by actively looking for CSR topics that one can contribute to. Answers and messages should be short and to the point and with an appropriate frequency, and in particular, no self-promotion may be added.

We found that recurring events, such as Earth Day or International Women's Day, must be considered within strategic communication as special content. In addition, attention should be paid to national or local grievances affecting society, so that personal charity projects can be initiated. Content-wise, it is thus communication on corporate giving such as philanthropic donation that increases social-media-engagement behavior, through the number of retweets, comments and likes. In addition, framing the message with a positive wording and being aware of positive trigger-words in the CSR context. such as 'environment', can add to positive stakeholder-perception and positive eWOM.

Our review results made clear that social CSR-topics are particularly suitable for generating positive effects on Twitter. Customers, for example, prefer engaging with posts on social issues which are aligned with the company's core purpose. It is thus crucial to not make sudden shifts due to trends or even topics other businesses are engaging with, because positive sentiment and a high level of engagement can demonstrably only be achieved if authentic topics that match the company's orientation and image are pursued over a longer period of time, and are built on one another.

Looking not only on content, but the tweet structure, tagging relevant stakeholders and key opinion leaders is also crucial when trying to trigger engagement as well as positive sentiments, not only about the tweet, but also the company itself. CSR topics need to be made explicit by the use of the right hashtags. Sometimes even a cause-related or unique company-cause-related hashtag can help with addressing the right community and differentiating the topic from more general ones within that cause. Finally, messages need to be diverse and of different levels of complexity, with videos being at the forefront.

### 5.2. Limitations and Future Studies

The presented review results are subject to a number of limitations. As studies included in the synthesis are diverse in geographic focus as well as the target groups selected, identified CSR-related subjects or even communication strategies may be less generalizable. Future research could address these limitations and perform comparative studies. The employed methods were not explicitly part of our review, which is why the quality of specifically ML methods could only be considered to a limited extent in the results discussion. Therefore, an analysis of the ML pipelines used in the context is another relevant field worth researching. Finally, we found that experimental research approaches in particular are often at odds with research findings based on original Twitter data. We therefore propose to repeat these approaches with Twitter data. Since it is not possible to predict how the use of Titter by companies on CSR topics will change after Elon Musk's takeover and the accompanying strategic realignments, this review provides the basis for a future comparative longitudinal study

### 5.3. Conclusions

With the increasing regulatory and social pressure on companies of all sectors in terms of authentic and transparent communication on CSR topics, an increase in digital communication via social media, and especially Twitter, can be observed over the past decade. Instead of the one-sided communication strategy initially adopted by marketing via the classic mass-media, the CSR-focus is increasingly shifting to dialogic communication as means to identify stakeholder needs. Companies' top management play a special role on social media, as CEOs have strong networks and thus opinion-shaping power, which can be used strategically to generate positive WOM. We were able to derive content focal-points

from the topics represented most frequently, which will prospectively increase in the next few years, both in quantity and content quality. In particular, social topics that directly affect, customers as well as local communities and employees generate interaction and thus profitable SMC, and strengthen loyalty or brand reputation. This connects to the main principles of stakeholder theory. The company should identify and then address the personal needs of stakeholders as far as possible. Companies need to become aware of social networks' advantages as CSR management tools and as drivers for collaborative interaction with stakeholders, which enable a more sustainable and inclusive implementation of CSR principles in their activities, if they are not perceived as greenwashing or social washing. In this context, companies achieve the best measurable results on social media when they incorporate long-standing corporate values into their CSR strategy, do not focus on trending topics, and when they act on the theory of legitimacy by engaging in proactive CSR communications rather than legitimizing negative outcomes for people and the environment, due to business activities. By directly addressing diverse stakeholders via mentions or tags, companies can further facilitate a platform for co-creation while they are at the same time broadcasting their social good deeds, which in a CSR context can lead to positive word-of-mouth and in the end create value for the company. It is striking that in not one study do ethical aspects play a significant role or are even mentioned regarding public Twitter-data-use, and therefore companies need to always consider the reflective, responsible, and ethically demanding use of social media, if they want to use the possibility of analyzing social media KPIs in the context of strategic CSR stakeholder management.

**Supplementary Materials:** The following supporting information can be downloaded at: https://www.mdpi.com/article/10.3390/su142416769/s1, Table S1: Sample.

**Author Contributions:** Conceptualization, K.P. and S.B.-J.; methodology K.P. and S.B.-J.; formal analysis, K.P.; writing—original draft preparation, K.P.; writing—review and editing, K.P.; visualization, K.P.; supervision, S.B.-J.; funding acquisition, S.B.-J. All authors have read and agreed to the published version of the manuscript.

**Funding:** This research was supported as part of the ATLAS project "Innovation and digital transformation in healthcare" by the state of North Rhine-Westphalia, Germany (grant number: ITG-1-1).

**Institutional Review Board Statement:** Not applicable.

**Informed Consent Statement:** Not applicable.

**Data Availability Statement:** Not applicable.

**Conflicts of Interest:** The authors declare no conflict of interest.

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
