# Peer review of "Corporate Social Responsibility on Twitter: A Review of Topics and Digital Communication Strategies’ Success Factors"

_sustainability, doi:10.3390/su142416769_

Round 1

Reviewer 1 Report

Dear Authors,

the idea behind the paper is original and could give an interesting contribution to the field. 

By the way, there are some necessary revisions:

·        In the introduction, the linkage between CSR and social media should be better developed.

·        At line 57, a source is needed.

·        Result and Discussion section, is sometimes really clear, sometimes a little confusing. This because you should move from a literature review to a bibliometric analysis, in order to reach the scientific standards required by the journal. I suggest to have a look to: Naveen Donthu, Satish Kumar, Debmalya Mukherjee, Nitesh Pandey, Weng Marc Lim, How to conduct a bibliometric analysis: An overview and guidelines, Journal of Business Research, Volume 133, 2021

Author Response

  1. In the introduction, the linkage between CSR and social media should be better developed.

Thank you for pointing this out, we further elaborated on this in the introduction:

In this context, trust, authenticity and transparent communication of information describe existing challenges of communication in social networks with respect to corporate social responsibility [9-11]. Successfully deployed social media can thus be a powerful tool for building sustainable corporate communication. To generate beneficial outcomes, compa-nies need to have answers to the following questions: On which CSR-related content should be communicated? Which stakeholders should be preferentially addressed? Which communication techniques shall be used to leverage stakeholder management outcomes (and in order not to be perceived as green or social washing)?

Busser, J. A.; Shulga, L. V., Involvement in consumer-generated advertising: Effects of organizational transparency and brand authenticity on loyalty and trust. International Journal of Contemporary Hospitality Management 2019.

Jiang, H.; Luo, Y., Crafting employee trust: from authenticity, transparency to engagement. Journal of Communication Management 2018.

Men, L. R.; Tsai, W.-H. S., Perceptual, attitudinal, and behavioral outcomes of organization–public engagement on corporate social networking sites. Journal of public relations research 2014, 26, (5), 417-435.

  1. At line 57, a source is needed.

Thank you, we added the Google Scholar URL within the text

  1. Result and Discussion section, is sometimes really clear, sometimes a little confusing. This because you should move from a literature review to a bibliometric analysis, in order to reach the scientific standards required by the journal. I suggest to have a look to: Naveen Donthu, Satish Kumar, Debmalya Mukherjee, Nitesh Pandey, Weng Marc Lim, How to conduct a bibliometric analysis: An overview and guidelines,Journal of Business Research, Volume 133, 2021

Thank you for this suggestion. We believe as stated in Donthu et al. that we did not analyze the social and structural relationships between different research constituents as the amount of literature is not large (42) and thus also not appropriate for a bibliometric analysis (e.g. to perform a network analysis). We stated this in the text and cited the suggested literature in this context. Since we performed a scoping review, we further elaborated on this, by adding arguments made by Munn et al. and added this citation: Munn, Z., Peters, M.D.J., Stern, C. et al. Systematic review or scoping review? Guidance for authors when choosing between a systematic or scoping review approach. BMC Med Res Methodol 18, 143 (2018). https://doi.org/10.1186/s12874-018-0611-x

Reviewer 2 Report

The paper has an interesting subject, but it needs some improvements as follows:

The section Results and Discussion should be interpretive rather than just descriptive and connect the research results with relevant literature citations for validity and reliability.

The figures and the tables need critical analysis.

We recommend seeing the following sources:

From MDPI: Burlea-Schiopoiu, A., Mihai, L.S. 2019. An Integrated Framework on the Sustainability of SMEs. Sustainability, 11(21), 6026; https://doi.org/10.3390/su11216026  

 From Springer: Burlea-Schiopoiu, A., Idowu, S., Vertigas, St., (editors), 2017, Corporate Social Responsibility in Times of Crisis: A Summary, Springer.

 Good luck!

Author Response

  1. The section Resultsand Discussion should be interpretive rather than just descriptive and connect the research results with relevant literature citations for validity and reliability.

Thank you for pointing this out. In this context we first made the underlying theories of legitimacy and stakeholder management more explicit in chapter 2 and discussed fininds additionally explcitely with the theoretical foundation when needed.

  1. The figures and the tables need critical analysis.

Thank you for pointing this out. You find the analysis and explainations to figure 1 in chapter “3.1. Data collection”. We rewrote the paragraph to make the connection more evident.

We further elaborated on figure two in chapter 4.1 Sample descriptives

We also added information on the analysis regarding Table 1. Target group descriptives.

  1. We recommend seeing the following sources:

From MDPI: Burlea-Schiopoiu, A., Mihai, L.S. 2019. An Integrated Framework on the Sustainability of SMEs. Sustainability, 11(21), 6026; https://doi.org/10.3390/su11216026  

From Springer: Burlea-Schiopoiu, A., Idowu, S., Vertigas, St., (editors), 2017, Corporate Social Responsibility in Times of Crisis: A SummarySpringer.

Thank you for your article suggestion. After reading the paper we find the research aim and scope not in line with our research as the aim of the recommended article is to investigate the relationship between the corporate social responsibility (CSR) budget, innovation and training, defined as sustainable factors, and the financial results of small and medium enterprises (SMEs). The research is conducted by analyzing the financial results of a sample of 200 SMEs from the southwestern region of Romania. The results show that SMEs can use training and innovation to improve the impact of CSR on their sustainability, focusing on positive financial indicators. The results show that corporate social responsibility (CSRBi), innovation (InnovBi) and training (TrainingBi) as sustainable factors are significantly and positively correlated with the following indicators: Profit (Profiti), Profit per Employee (ProfitEi) and Total Expenses (Expensesi), and negatively correlated with Debt Ratio (DebtRi).

Reviewer 3 Report

Dear authors

Congratulations on the theme developed in your manuscript. However, to strengthen the document, the following is recommended:

1. Improve the introduction, adding the challenges of communication in social networks with respect to corporate social responsibility.

2. Add within the theoretical foundation the relationship between marketing, social media and social responsibility within the framework of the theory of Stakeholders and Digital Marketing (Advertising in social media). Verify if the theory of legitimacy can be part of the theoretical foundation of the work.

3. Justify why the use of the PRISMA software for the development of the meta-analysis and why not use other software such as: Vosviewer, SciMAT. State the reasons why the Scopus and/or Web of Science databases were not used.

4. It is recommended that after the discussion of the results the conclusions be incorporated, which will have to be improved and expanded. This section should include how the results coincide or do not coincide with the theories underlying the research.

Regards

Author Response

  1. Improve the introduction, adding the challenges of communication in social networks with respect to corporate social responsibility.

Thank you for pointing this out, we further elaborated on this in the introduction:

In this context, trust, authenticity and transparent communication of information describe existing challenges of communication in social networks with respect to corporate social responsibility. Successfully deployed social media can thus be a powerful tool for building sustainable corporate communication. To generate beneficial outcomes, companies need to have answers to the following questions: On which CSR-related content should be communicated? Which stakeholders should be preferentially addressed? Which commu-nication techniques shall be used to leverage stakeholder management outcomes?

Busser, J. A.; Shulga, L. V., Involvement in consumer-generated advertising: Effects of organizational transparency and brand authenticity on loyalty and trust. International Journal of Contemporary Hospitality Management 2019.

Jiang, H.; Luo, Y., Crafting employee trust: from authenticity, transparency to engagement. Journal of Communication Management 2018.

Men, L. R.; Tsai, W.-H. S., Perceptual, attitudinal, and behavioral outcomes of organization–public engagement on corporate social networking sites. Journal of public relations research 2014, 26, (5), 417-435.

  1. Add within the theoretical foundation the relationship between marketing, social media and social responsibility within the framework of the theory of Stakeholders and Digital Marketing (Advertising in social media). Verify if the theory of legitimacy can be part of the theoretical foundation of the work.

Thank you for pointing this out. We stated in our introduction that … to enhance stakeholders’ company perception, its actions and ethics [3], CSR disclosure is becoming a more sophisticated and strategically motivated process with expectations of a return [4]. Thus, carrying out socially responsible behavior becomes a strategy of legitimation and survival – as we strongly believe that the theory of legitimacy is already part of the general theoretical foundation. As chapter two picks up the subject of “Digital CSR communication strategies and indicators of success” we added an elaboration of stakeholder theory and the theory of legitimacy to make the connection more explicit and renamed the chapter to: The role of digital CSR communication strategies for stakeholder-management

  1. Justify why the use of the PRISMA software for the development of the meta-analysis and why not use other software such as: Vosviewer, SciMAT.

We performed a scoping review and for sample selection we used the PRIMSA flow chart , which is not a software but a systematic approach to transparent literature selection. As part of the data analysis, we chose a human (qualitative) approach rather than a software to perform a network analysis, which could be the focus of another paper.

  1. State the reasons why the Scopus and/or Web of Science databases were not used.

Thank you for this question. Research by Martin-Martin et al. (2019) comparing GoogleScholar, Web of Science and Scopus shows, that a significant amount of literature in the Social Sciences and Humanities were not covered by the selective databases Web of Science or Scopus. In most cases, the cause was that the database did not cover the journal at the time the article was published. We thus chose to use GoogleScholar. For further insights:

Martín-Martín, A., Orduna-Malea, E., Thelwall, M., & Delgado-López-Cózar, E. (2019). Google Scholar, Web of Science, and Scopus: which is best for me?. Impact of Social Sciences Blog.

  1. It is recommended that after the discussion of the results the conclusions be incorporated, which will have to be improved and expanded. This section should include how the results coincide or do not coincide with the theories underlying the research.

Thank you for pointing this out, we expanded the conclusion.